# Shame-Based Experiences of Homophobic Bullying and Mental Health: The Mediating Role of Self-Compassionate Actions

**DOI:** 10.3390/ijerph192416866

**Published:** 2022-12-15

**Authors:** Daniel Seabra, Jorge Gato, Diogo Carreiras, Nicola Petrocchi, Maria do Céu Salvador

**Affiliations:** 1University of Coimbra, Center for Research in Neuropsychology and Cognitive and Behavioural Intervention (CINEICC), Rua do Colégio Novo, 3000-115 Coimbra, Portugal; 2Faculty of Psychology and Education Sciences and Center for Psychology, University of Porto, Rua Alfredo Allen, 4200-135 Porto, Portugal; 3Department of Economics and Social Sciences, John Cabot University, Via della Lungara 233, 00165 Rome, Italy; 4Compassionate Mind Italia, Via Marcantonio Colonna 44, 00192 Rome, Italy

**Keywords:** sexual minorities, traumatic shame experiences, homophobic bullying, psychopathology, self-compassionate actions

## Abstract

Homophobic experiences with traumatic characteristics related to shame are more frequent among sexual minority (SM) than heterosexual individuals. Concurrently, SM individuals present higher levels of psychopathology and transdiagnostic processes (e.g., shame) than heterosexual individuals. Self-compassion has been identified as a protective mechanism that counteracts the effects of shame. The current study aimed to analyse which components of self-compassion affect mental health and test the mediating role of self-compassion in the relationship between feelings of shame in traumatic homophobic bullying experiences (THBEs) and psychopathology indicators (depression, anxiety, and social anxiety symptoms). In this study, 190 Portuguese SM individuals (*M*age = 28.3, *SD* = 7.5) completed self-report measures assessing traumatic experiences, self-compassion, and psychopathology. Data were explored with SPSS and AMOS. Regression analyses showed that internal shame felt during THBE and compassionate actions predicted psychopathology outcomes. Mediation analyses revealed that internal shame during a THBE had a significant indirect effect on all psychopathology outcomes through compassionate actions. In other words, internal shame during a THBE was significantly associated with depression, anxiety, and social anxiety, and these relationships were partially mediated by compassionate actions. Our results reinforce the importance of developing compassionate actions towards the self as a possible protective factor for psychopathology among SM individuals.

## 1. Introduction

In recent decades, transformations and transitions in society have allowed further understanding of sexual diversity [1]. Nonetheless, most societies still deny and stigmatize sexual minority (SM) individuals [2,3]. Stigmatized SM individuals, including lesbian, gay, bisexual, pansexual, asexual, and other non-heterosexual people, are more likely to present increased levels of mental health disorders [4,5], particularly anxiety and depression [6], and social anxiety symptoms [7] in comparison to heterosexual individuals. This increased risk for adverse mental health outcomes is explained by minority stress, which includes four stigma processes on a continuous spectrum from distal to proximal [3]. Distal stressors are events and experiences outside the person [8], for example, violence and discrimination [3]. In the school context, SM adolescents are often targets of homophobic bullying [9], which is characterized by verbal or physical acts of harassment related to one’s perception of sexual orientation due to behaviours that fall outside the heteronormative framework [10]. These acts occur when someone is perceived as not having a heterosexual sexual orientation, a cisgender gender identity, or a non-conforming gender expression [11], and thus heterosexual and cisgender people can also be victims of homophobic bullying [12,13,14]. Some authors have found that early shame experiences may involve traumatic memory characteristics [15], with a lasting negative impact on mental health [15,16]. Memories of early shaming homophobic events may, therefore, also have traumatic characteristics, and victims of these events may often resort to shame-avoidance strategies [17].

Gilbert [18] defined compassion as “being open to the suffering of self and others, in a non-defensive and non-judgmental way. Compassion also involves a desire to relieve suffering, cognitions related to understanding the causes of suffering, and behaviours–acting with compassion” (p. 1). Thus, three different flows of compassion have been described: people can feel compassion for others, receive compassion from others, and be compassionate towards themselves [19]. The third flow, also called self-compassion, involves an affective state of sensitivity to own suffering with a commitment to try to alleviate and prevent it [20]. In fact, self-compassion is a motivational response to own experience of suffering with two crucial processes: motivation to engage with suffering and motivation to take helpful action to relieve it [21]. Self-compassion has been identified by several studies as a protective mechanism for anxiety and depression [22] and social anxiety [23]. It also seems particularly helpful to counteract high levels of self-criticism and shame [24,25] among SM individuals [26,27]. Accordingly, self-compassion has also been indicated as an important psychological process to improve well-being and resilience [28,29] and attenuate the impact of stress, anxiety, depression, and suicidality in SM individuals [30]. Additionally, self-compassion significantly predicts life satisfaction in gay men [31]. In this regard, Matos and colleagues [32] underlined the importance of developing self-compassion towards memories of adverse early life events among gay men who had experienced shame and depression.

The current study aimed to: (a) examine if both self-compassion components (compassionate engagement and actions) predicted SM individuals’ mental health (anxiety, depression, and social anxiety symptoms), and (b) analyse the mediating effect of self-compassion in the relationship between feelings of (external and internal) shame during a THBE and mental health indicators (anxiety, depression, and social anxiety symptoms) among SM individuals. We hypothesized that both self-compassion components would have a significant effect on mental health indicators and that both compassionate engagement and actions would mediate the relationship between shame during a THBE and mental health indicators. The proposed mediation model is presented in Figure 1.

## 2. Materials and Methods

### 2.1. Participants

The sample was composed of 190 SM adults who reported having had traumatic experiences based on their sexual orientation and/or gender expression. Participants were recruited using different methods: 3.20% through newsletters of non-governmental organizations (NGOs), 8.90% through the snowball technique, 31.60% through convenience sampling (direct contact of researchers), and 55.30% through social networks (e.g., *Facebook*). All participants were Portuguese, and most were cisgender, single, and did not have children. The sample characteristics are described in Table 1.

### 2.2. Procedures

Data for the present cross-sectional study were collected between January and March 2020 and were collected using a web-based survey in the context of larger research. Confidentiality and voluntary participation were assured. After reading a page with the study information, participants gave their free and informed consent and completed the research protocol. Inclusion criteria were self-identification as a sexual minority, being Portuguese, age between 18 and 65 years old, and full completion of the questionnaires. There was no financial compensation for the participation.

### 2.3. Measures

*Sociodemographic Information*. Participants were asked about sociodemographic characteristics, such as age, gender, gender identity, sexual orientation, marital status, if they had children, educational level, employment status, and if they were receiving psychological treatment at the time of the study. Sociodemographic information is described in Table 1.

*Trauma-Related Shame Inventory (TRSI).* Originally from Øktedalen and colleagues [33] and with a European Portuguese version from Cid and Pinto-Gouveia [34], this scale has 24 items to assess the negative evaluation of the self in the context of trauma with a painful affective experience and a behavioural tendency to hide and withdraw from others to conceal one’s own perceived deficiencies. Each subscale has 12 items, and Cronbach’s alphas of .95 and .94 for external shame (e.g., “If others knew what happened to me, they would be ashamed of me”) and internal shame (e.g., “I am ashamed of the way I felt during my traumatic experience”) were reported in the European Portuguese version [34]. Items are rated on a 4-point Likert scale from *Not true of me* (0) to *Completely true of me* (3). Higher mean scores indicate higher levels of shame felt in the traumatic experience. In our sample, Cronbach’s alphas were .95 for both subscales.

*The Compassionate Engagement and Action Scales (CEAS).* From Gilbert and colleagues [20], this scale assesses three different flows of compassion: compassion for others, compassion from others, and compassion toward the self. Each section has eight items about competencies that facilitate turning towards and engaging in suffering (compassionate engagement) and another five items about competencies that facilitate actions to alleviate and prevent suffering (compassionate actions). In this study, only the subscale compassion toward the self was used (e.g., “I am motivated to engage and work with my distress when it arises” and “I direct my attention to what is likely to be helpful to me”). Participants report their answers on a 10-point Likert scale from *Never* (1) to *Always* (10). Higher mean scores indicate higher levels of self-compassion. In our sample, the subscale compassion toward the self presented good reliability (Cronbach’s alpha of .79).

*Depression, Anxiety and Stress Scales, 21-Item Version (DASS-21)*. Originally from Lovibond and Lovibond [35] and with a European Portuguese version from Pais-Ribeiro and colleagues [36], this scale has 21 items divided into three subscales: depression, anxiety, and stress symptoms, with Cronbach’s alphas between .74 and .81 in the validation study. Items are rated on a 4-point Likert scale from *Did not apply to me at all* (0) to *Applied to me very much or most of the time* (3), with higher scores indicating greater negative affect. In this study, only the anxiety (physical arousal symptoms, panic attacks, and fear, e.g., “I was aware of the action of my heart in the absence of physical exertion”) and depression symptoms (symptoms usually associated with negative mood, e.g., “I could see nothing in the future to be hopeful about”) subscales were used. Cronbach’s alphas were .90 and .91 for anxiety and depression symptoms, respectively.

*Social Interaction Anxiety Scale (SIAS).* Originally from Mattick and Clarke [37] and with a European Portuguese version from Pinto-Gouveia and Salvador [38], this scale has 19 items and assesses fears of general social interaction (e.g., “When mixing socially, I am uncomfortable”). Items are rated on a 5-point Likert scale from *Not at all characteristic or true of me* (0) to *Extremely characteristic or true of me* (3), with higher total scores indicating higher levels of social anxiety. Cronbach’s alpha in this study was .92.

### 2.4. Statistical Analyses

All data analyses were conducted in IBM Statistical Package for the Social Sciences version 27 (SPSS (IBM Inc., Chicago, IL, USA)) [39] and IBM AMOS version 27 (IBM Inc., Chicago, IL, USA) [40]. The normality of data distribution was examined using Kolmogorov–Smirnov test and deviations through skewness (*Sk*) and kurtosis (*Ku*) values. Only values above |3|, |10| for *Sk* and *Ku*, respectively, were considered to represent severe violations of normal distribution [41,42]. Pearson’s correlation coefficients (*r*) were used to examine the association between variables and to examine mean differences; Student’s *t*-tests (*t*) and Mann–Whitney U tests (*U*) were used as parametric and nonparametric analyses, respectively. The correlations were interpreted according to Dancey and Reidy [43]: correlation below .30 means a weak association, between .40 and .60 means a moderate association, and above .70 means a strong association. For examination of effect sizes of specific mean differences, Cohen’s *d* (.2 = small effect, .5 = medium effect, and .8 = large effect) [44] in parametric analyses and Rosenthal’s *r* for nonparametric analyses (.1 = small effect, .3 medium effect, and .5 large effect) [44,45] were used. In the hierarchical regressions, the independence of errors was analysed by the Durbin–Watson test, with acceptable values between 1 and 3 [45]. Multicollinearity was examined through variance inflation factor (VIF), whose values should be inferior to 10 [45]. Regarding model fit, the indices ascertained were chi-squared (*χ*^2^), comparative fit index (CFI), Tucker–Lewis index (TLI), and root mean square error of approximation (RMSEA). Chi-squared (*χ*^2^) should be nonsignificant, and, for comparative fit indexes, higher values indicate a better fit [46]: fit indexes values between .80 and .89 are poor [42] and between .90 and .95 reflect a good fit [42,47]. Lower RMSEA values indicate better fit [42]. A bootstrap procedure (10,000) was performed with a confidence interval of 95%. Significance was considered when intervals did not include zero. The AMOS provides values for total, direct, and total indirect effects. To analyse specific indirect effects, the researchers calculated the product of the direct effect of the independent variable on the mediator variable and the total effect of mediator variables on the dependent variable [48].

## 3. Results

### 3.1. Normality, Descriptive Statistics, and Correlations between Variables

All the variables except compassionate engagement had a non-significant Kolmogorov–Smirnov value but presented a *Sk* and *Ku* < 2. That is, no severe violations to normality were found. The VIF’ values were all below 4. Descriptive statistics and Pearson’s correlation coefficients are presented in Table 2. Feelings of (external and internal) shame in THBE and mental health indicators (anxiety, depression, and social anxiety symptoms) showed significant, positive, and moderate correlations (.40 < *r* < .59; *p* < .001); that is, higher levels of shame in THBEs were associated with higher levels of mental health indicators. Self-compassion showed negative and weak to moderate associations to internal and external shame (−.17 < *r* < .40, *p* < .05), as well as with anxiety and depression symptoms (−.15 < *r* < .40, *p* < .001). Social anxiety symptoms also revealed significant, negative, and weak to moderate correlations with self-compassion’s total score (*r* = −.31, *p* < .001) and with compassionate actions (*r* = −.37, *p* < .001). The correlation between social anxiety symptoms and compassionate engagement was nonsignificant (*r* = −.14, *p* = .057). All other correlations with compassionate engagement were negative and significant but weak, and correlations of compassionate actions with all other variables were among the strongest ones. The strongest correlations were found between depression symptoms and (i) THBE internal shame (*r* = .59, *p* < .001) and (ii) compassionate actions (*r* = .50, *p* < .001). In general, higher levels of self-compassion were associated with lower levels of shame felt in THBE, anxiety, depression, and social anxiety symptoms.

### 3.2. Regression Analyses

Differences as a function of gender, gender identity, and sexual orientation were analysed to explore the need to introduce covariates in the regression models. Considering the small sample size of some groups (e.g., non-binary, gender identity other, asexual people), differences were tested only for (i) participants who identified as women vs. men, (ii) participants who identified as cisgender vs. transgender people, and (iii) participants who identified as monosexual (gay men and lesbian women) vs. bi+ (bisexual and pansexual individuals). Regarding gender, women presented higher levels of internal shame felt in THBE (*t*_(73)_ = 2.74, *p* = .008, *d* = .48) and depression symptoms (*t*_(173)_ = 2.34, *p* = .020, *d* = .39), as well as lower levels of self-compassion (*t*_(173)_ = −2.16, *p* = .032, *d* = .35) and compassionate actions (*t*_(173)_ = −2.47, *p* = .015, *d* = .41), when compared to men. Transgender people presented higher levels of external shame felt in THBE (*U* = 577.50, *p* = .009, *r* = −.19), anxiety symptoms (*U* = 477, *p* = .002, *r* = −.23), and depression symptoms (*U* = 507.50, *p* = .003, *r* = −.22) when compared to cisgender people. Differences in sexual orientation were found both in internal (*t*_(86)_ = −3.28, *p* < .001, *d* = .55) and external (*t*_(87)_ = −2.99, *p* = .004, *d* = .50) shame felt in THBE, anxiety (*t*_(83)_ = −2.72, *p* = .008, *d* = .46), and depression symptoms (*t*_(185)_ = −2.32, *p* = .022, *d* = .37), bi+ individuals presenting higher levels in these variables when compared to monosexual individuals. Parametric mean comparisons (gender and sexual orientation) are presented in Table A1, and non-parametric mean comparisons (gender identity) are presented in Table A2. Both tables are available in Appendix A.

Hierarchical regression models predicting anxiety, depression, and social anxiety symptoms were performed. Considering the abovementioned differences in gender, gender identity, and sexual orientation, these variables were included in the regression model to control its potential confounding effect. The first block of predictors included gender identity (0 = cisgender; 1 = transgender) and sexual orientation (0 = monosexual; 1 = bi+) for anxiety; and gender (0 = women; 1 = men), gender identity, and sexual orientation for depression. The second block comprised the dimensions of (external and internal) shame felt in THBE, and the third block included the dimensions of self-compassion: compassionate engagement and compassionate actions. All models were significant and explained 29.70% of anxiety, 46.10% of depression, and 24.50% of social anxiety symptoms variance. Estimates and standard errors of regressions are detailed in Table 3. In the last model, only internal shame felt in THBE and compassionate actions predicted mental health indicators (internal shame with a positive prediction and compassionate actions with a negative prediction); gender identity also predicted anxiety and depression symptoms, and compassionate engagement positively predicted depression symptoms.

### 3.3. The Mediating Effect of Self-Compassion in the Relationship between Shame Felt in THBE and Mental Health Indicators

Considering the regression results, the initial mediation model had internal shame felt in THBE as an independent variable and anxiety, depression, and social anxiety symptoms as dependent variables. Additionally, compassionate actions were used as a mediator for all dependent variables, and compassionate engagement was used as a mediator only for depression symptoms. Gender identity was controlled for anxiety and depression symptoms. The final model, adjusted to 183 individuals, is illustrated in Figure 2 and had twenty-two parameters and six degrees of freedom.

The model showed an excellent fit, *χ*^2^ _(6)_ = 5.44, *p* = .489, CFI = 1.00, TLI = 1.01, RMSEA < .001. All paths were statistically significant. Altogether, predictors explained 31% of anxiety symptoms, 45% of depression symptoms, and 26% of social anxiety symptoms’ variance.

Before introduction of mediator variables, the total effects were significant: β_Anxiety.InternalShame_ = .467; *p* < .001; 95% CI [.322; .596], β_Depresssion.InternalShame_ = .579; *p* < .001; 95% CI [.472; .672], and β_SocialAnxiety.InternalShame_ = .470; *p* < .001; 95% CI [.353; .575]. After introduction of the mediators, the model presented significant direct effects: β_Anxiety.InternalShame_ = .379, *p* < .001; 95% CI [.222, .515]; β_Depression.InternalShame_ = .444, *p* < .001; 95% CI [.327, .552]; and β_SocialAnxiety.InternalShame_ = .379, *p* < .001; 95% CI [.252, .501]. The indirect effect of internal shame felt in THBE through compassionate actions on anxiety symptoms (β_Anxiety.InternalShame|Actions_ = .091; *p* < .001; 95% CI [.036, .158]) and on social anxiety symptoms (β_SocialAnxiety.InternalShame|Actions_ = .091; *p* = .005; 95% CI [.024, .164]) was significant. The total indirect effect of internal shame felt in THBE through compassionate engagement and actions on depression symptoms was also significant (β_Depression.InternalShame|Actions.Engagement_ = .135; *p* < .001; 95% CI [.074, .211]). The indirect effect through compassionate actions corresponded to 4.25% of the total effect of internal shame felt in THBE on anxiety symptoms (.091 × .467 = .0425) and 4.28% on social anxiety symptoms (.091 × .470 = .0428).

The total indirect effect mediated by compassionate engagement and actions corresponded to 7.82% of the total effect of internal shame felt in THBE on depression symptoms (.135 × .579 = .0782). Specifically, 2.29% of the specific indirect effect of THBE on depression symptoms goes through compassionate engagement and 79.50% through actions. These findings revealed that compassionate actions had a significant indirect effect on the relationship between internal shame felt in THBE and anxiety and social anxiety symptoms. Moreover, compassionate engagement and actions had a significant indirect effect on the relationship between internal shame felt in THBE and depression symptoms (compassionate actions had a higher indirect effect when compared to compassionate engagement). Overall, the model suggested that compassionate actions is a partial positive mediator of the relationship between internal shame felt in THBE and anxiety and social anxiety symptoms, and compassionate engagement and actions are a partial mediator of the relationship between internal shame felt in THBE and depression (compassionate engagement was a positive mediator and compassionate actions was a negative mediator).

## 4. Discussion

The present study had two aims: (a) to examine which self-compassion components might have a significant effect on sexual minorities’ mental health, and (b) to analyse the mediating effect of self-compassion in the relationship between feelings of shame in THBE and mental health indicators among SM individuals.

All the associations between external and internal shame felt in THBE, self-compassion and its components, anxiety, depression, and social anxiety symptoms were significant except the association between compassionate engagement and social anxiety. Self-compassion and its components presented a negative correlation with all the remaining variables; that is, higher levels of self-compassion were associated with lower levels of (external and internal) shame felt in THBE, anxiety, and with depression and social anxiety symptoms. A negative association of self-compassion and shame with psychopathology indicators has been found and discussed in previous studies that did not focus on sexual orientation [49,50]. Our results extend these findings to SM individuals, reinforcing that self-compassion might be a transversal positive factor to buffer shame and psychopathology indicators across sexual orientations.

In comparative analyses, there were some differences in shame felt in THBE, psychopathology indicators, and self-compassion when considering gender, gender identity, and sexual orientation. Higher levels of internal shame felt in THBEs were found in women when compared to men. In fact, the risk to develop PTSD is higher in women [51], and other studies have also found that women presented higher levels of internal shame when compared to men [52]. Ferguson and Eyre [53] clarified that gender-related stereotypes encourage women to make greater shame-relevant appraisals than men. Considering the evolutionary and biopsychosocial model for shame [54], internal shame is associated with self-devaluation and internal attributions (including bullying experiences). Therefore—and considering Western gender stereotypes—our results suggest that women who are victims of bullying experiences may engage more in internal and shaming appraisals and attributions. Additionally, women presented increased levels of depression when compared to men. This result is in line with many other studies [55,56]. No gender differences were found in anxiety and social anxiety symptoms.

Regarding gender identity, transgender individuals showed higher levels of external shame felt in THBE than cisgender individuals. This result might be related to the fact that transgender individuals present more experiences of victimization when compared with cisgender individuals [57]. Furthermore, considering the connection between shame and stigma (more prevalent among transgender individuals), the shame of transgender individuals might be associated with the heavy social stigma they endure [2]. Bockting and colleagues [58] found that shame (along with pride, passing, and alienation) was also one of the components of internalized transphobia. Considering the results of this study, transgender individuals seem to have a higher traumatic shame effect focused on others’ evaluations (external shame), reinforcing the connection between stigma, external shame, and impact on mental health.

Transgender individuals also presented higher levels of anxiety and depression symptoms when compared to cisgender individuals, in line with Borgogna and colleagues’ [4] study. These results might be explained by the minority stress model. This population must face more specific challenges and additional stress compared to cisgender individuals, such as misgendering or non-affirmative, legal, and medical care issues [59,60]. No gender identity differences were found for social anxiety symptoms. This result was unexpected. In fact, studies showed that transgender individuals presented higher levels of social anxiety when compared to cisgender individuals [61]. As the sample of transgender individuals in this study is small (*n* = 12), our result may not reflect reality; more reliable findings would perhaps be found using a wider sample.

Regarding sexual orientation, higher levels of both external and internal shame felt in THBEs were found in bi+ individuals when compared to monosexual individuals. Hequembourg and Dearing [62] found no differences in shame-proneness between sexual orientations. However, the frequency of sexual-minority-based prejudice experiences in bisexual people was associated with shame (and not with rejection sensitivity, as in gay men and lesbian women) [63]. In fact, bisexual men and women tend to report higher levels of stigma than homosexual men and women [2,64]. These results suggest that homophobic bullying experiences may have a higher shaming impact on bi+ individuals, probably due to the invisibility and stigma associated with this sexual orientation, also known as binegativity, bisexual erasure, or bisexual marginalization [2]. Bi+ also presented higher levels of anxiety and depression symptoms when compared to monosexual individuals in this study, in line with Ross and colleagues’ systematic review [65]. No sexual orientation differences were found for social anxiety symptoms. Similar to gender identity, this result was also unexpected. In other studies, Bi+ have presented higher levels of social anxiety when compared to monosexual individuals [7]. We hypothesise that, because THBEs are directed at an identity feature, regardless of sexual orientation, social anxiety that may derive from these experiences will not differentiate individuals that suffered from it.

Finally, in self-compassion, the only difference found was across genders. Women showed lower levels of self-compassion when compared to men, as other studies pointed out [66]. Specifically, women presented lower levels of self-compassionate actions when compared to men. These lower levels of compassionate actions to alleviate or prevent suffering may help to understand the higher levels of depression of women compared to men, as described above. Additionally, the traditional gender roles and social power structures allow women to be soft, nurturing, and tender but, at the same time, discourage them to be fierce, determined, and strong [67]. Our results mirror that: men have higher levels of compassionate actions, which might be reinforced by masculine culture. This masculine expected fierceness and strength does not necessarily include the caring tone of compassion.

The results from linear regressions were unexpected considering our initial hypothesis. The data showed that external shame felt in THBE did not present a significant effect on psychopathological symptoms. External shame is related to fears and beliefs about the social world, that is, one’s perception of being negatively viewed by others [54,68]. It seems that, for SM individuals with THBE, the perception that others have negative appraisals of them seems to have no effect on psychopathological outcomes. Nonetheless, their perception of own inadequacy and inferiority (i.e., internal shame) [54] had a unique effect on the same outcomes when included in the same model with external shame. These findings suggest that feeling ashamed of oneself, regardless of what others could think, contributes to the existence of mental health issues in SM individuals with THBE. Considering that our sample was recruited through an online survey, it is also possible that individuals that participate in these types of recruitment have lower levels of concealment of their sexual orientation. In this sense, perhaps individuals with a SM sexual orientation that respond to these surveys may not be very concerned about what others may think. Nevertheless, the experience of internal shame related to past THBE seems to linger and impact current mental health. The initial hypothesis also considered that both components of self-compassion (compassionate engagement and actions) would negatively predict mental health indicators. Surprisingly only compassionate actions negatively predicted all mental health indicators (anxiety, depression, and social anxiety symptoms). These results suggest that skills that facilitate compassionate actions to alleviate and prevent suffering are the most important factor to cope with anxiety, depression, and social anxiety symptoms and that compassionate engagement is not enough to have such an impact. Unexpectedly, compassionate engagement positively predicted depression symptoms. According to compassion-focused therapy [19,54], it is necessary to identify, note, and embrace suffering (compassionate engagement) to allow to choose effective actions for relieving suffering (compassionate actions). We hypothesize that, when SM individuals only note and embrace suffering (compassionate engagement without compassionate actions for relieving), they can develop depression symptoms. Additionally, anxious participants may have interpreted compassionate items as withdrawal or avoidance, which may have decreased anxiety symptoms and bypassed compassionate engagement. Involvement with suffering, *per se*, increases depression symptoms. This is also in line with a study by Di Bello and colleagues [69] that found an increase in sadness after visualization of a video that stimulates empathic sensitivity (compassionate engagement) and only a decrease after visualization of a video that stimulates compassionate actions.

The mediation analyses reinforced the potentially protective role of self-compassion, namely compassionate actions for anxiety, depression, and social anxiety symptoms in SM individuals who had THBE. The mediation model presented good fit indexes and explained 45% of depression, 31% of anxiety, and 26% of social anxiety symptoms. Internal shame felt in THBE showed a significant indirect effect through compassionate actions in psychopathology indicators in SM individuals. In other words, for this population, shaming traumatic homophobic experiences seem to impact psychopathology partially through the lack of compassionate actions. In fact, self-compassion has been indicated as a positive psychological process for well-being in this population [70,71,72]. These results seem consistent with other studies that presented negative and moderate correlations between shame and self-compassion [50], including other minority populations (e.g., people living with HIV) [73].

There is evidence that young people, compared to older individuals, report higher levels of insecurity and discomfort related to homophobic experiences (36.8% due to sexual orientation and 27.9% due to gender expression) [74]. These experiences often configure experiences of shame, and some of them may have traumatic characteristics [15]. Moreover, for sexual minority and gender diverse individuals, shame-based traumatic experiences have a direct impact on mental and physical health [75]. The results of the present study indicate that it is the internal shame felt in THBE (and not external shame) that is associated with the psychopathological symptoms among SM individuals. Additionally, compassionate actions seem to play an important protective role in comprehension of these symptoms. On the one hand, feelings of internal shame may block individuals from showing their true selves [76], leading them to conceal their sexual orientation and develop internalized stigma, which are proximal processes of minority stress responsible for poorer levels of mental health [3]. On the other hand, a self-compassionate posture can counteract shame and protect against psychopathology. This competency may be cultivated through compassionate mind training, leading to development of internal warmth, safeness, and soothing experiences [77].

Some limitations of this study should be considered. A more balanced and representative sample of the different SM orientations would allow for more solid generalization of results. All the measures were obtained through self-report questionnaires, and other ways to assess these variables would be valuable. In particular, the THBEs are self-reported and depend on a subjective report. Future studies should thus complement assessment of THBEs with a qualitative interview. Other positive factors, whether psychological (e.g., resilience, mindfulness, psychological flexibility) and/or social processes (e.g., social safeness and connection with the community) may also be explored in upcoming works.

This study has highlighted the role of internal shame and a negative self-perspective about oneself during THBEs. Moreover, compassionate actions have an important positive role against anxiety, depression, and social anxiety symptoms. Regarding clinical implications, internal shame should be a target in clinical practice with this population; also, considering the positive role of compassionate actions, this process may represent a valuable asset if included in clinical interventions with SM individuals. Additionally, self-compassion is associated with resilience in several contexts [78,79,80]. Resilience is also a protective factor for SM [8], and, as with self-compassion, resilience contributes to psychological well-being in different ways [81]. Both should be considered together in interventions with SM individuals.

## Figures and Tables

**Figure 1 ijerph-19-16866-f001:**
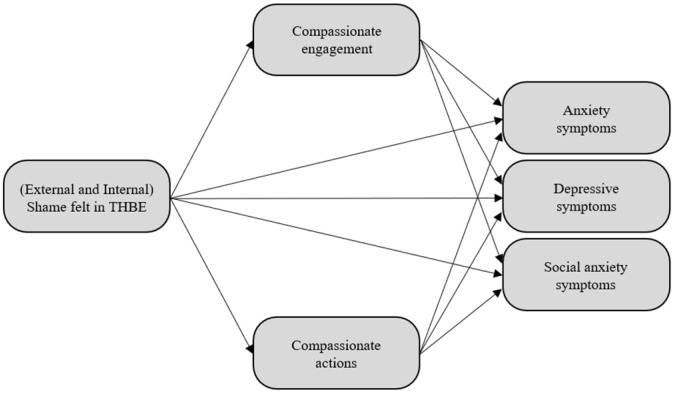
The proposed model in which both internal and external shame felt in traumatic homophobic bullying experience indirectly predict mental health indicators (anxiety, depression, and social anxiety symptoms) through both compassionate engagement and actions.

**Figure 2 ijerph-19-16866-f002:**
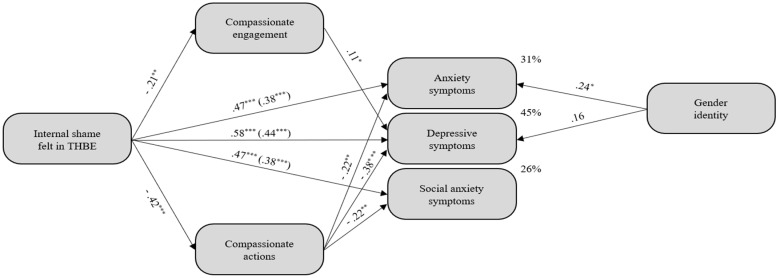
The mediation effect of engagement and actions in the relationship between internal shame felt in traumatic homophobic experience and anxiety, depression, and social anxiety symptoms. Standardized path coefficients among variables are presented *(n =* 186). This figure is simplified. The mediators and dependent variables have associated errors that are not presented. * *p* < .05; ** *p* < .005; *** *p* < .001.

**Table 1 ijerph-19-16866-t001:** Participants’ sociodemographic characteristics.

Characteristic	Sample (*N* = 190)	
	*n*	%
Gender		
Woman	54	28.4
Man	121	63.7
Non-binary	13	6.8
Other	2	1.1
Gender identity		
Cisgender	171	90
Transgender	12	6.3
Other	7	3.7
Sexual orientation		
Lesbian	27	14.2
Gay	105	55.3
Bisexual	39	20.5
Pansexual	16	8.4
Asexual	1	.5
Marital status		
Single	171	90
Married/living together as a couple	16	8.4
Divorced	3	1.6
Children ^a^	7	3.4
Educational level		
(until) Middle school	1	.5
Intermediate school	35	18.4
Graduate	69	36.3
Master	78	41.4
PhD	5	2.6
Post-PhD	2	11
Employment		
Unemployed	16	8.4
Student	44	23.2
Part-time employed	93	48.9
Full-time employed	17	8.9
Working student	20	10.5
Previous psychological treatment ^a^	40	21.1

Participants were on average 28.3 years (*SD* = 7.5); ^a^ reflects the number and percentage of participants answering “yes” to this question.

**Table 2 ijerph-19-16866-t002:** Descriptive statistics and correlations between study variables (*N* = 190).

Variable	*M (SD)*	1	2	3	4	5	6	7	8
1. External shame in THBE	1.4 (.6)	−1							
2. Internal shame in THBE	1.6 (.7)	.84 **	1						
3. Self-compassion (total)	6.6 (1.6)	−.29 **	−.36 **	1					
4. Compassionate engagement	6.5 (1.5)	−.17 **	−.20 **	.83 **	1				
5. Compassionate actions	6.7 (2.1)	−.31 **	−.40 **	.91 **	.53 **	1			
6. Anxiety symptoms	4.3 (4.8)	.45 **	.48 **	−.33 **	−.16 *	.39 **	1		
7. Depressive symptoms	6.1 (5.3)	.49 **	.59 **	−.40 **	−.15 *	−.50 **	.71 *	1	
8. Social anxiety symptoms	32.9 (18.2)	.40 **	.47 **	−.31 **	−.14	−.37 *	.44 *	.47 *	1

* *p* < .05; ** *p* < .001.

**Table 3 ijerph-19-16866-t003:** Hierarchical regression results for anxiety, depression, and social anxiety (*n* = 181).

Variable	Anxiety Symptoms	Depression Symptoms	Social Anxiety Symptoms
	*B (SE)*	*B (SE)*	*B (SE)*
Step 1			
Gender	N/A	**−2.18 * (.91)**	N/A
Gender identity	**3.56 * (1.57)**	**6.98 ** (2.21)**	N/A
Sexual orientation	**1.63 * (.78)**	.35 (.96)	N/A
Step 2			
Gender	N/A	−1.03 (.77)	N/A
Gender identity	**3.52 * (1.40)**	**5.72 ** (1.86)**	N/A
Sexual orientation	.46 (.72)	−.90 (.82)	N/A
External shame in THBE	.90 (.98)	−.03 (1.06)	1.06 (3.81)
Internal shame in THBE	**2.55 ** (.82)**	**4.38 *** (.88)**	**12.04 *** (3.21)**
Step 2			
Gender	N/A	−.52 (.71)	N/A
Gender identity	**3.40 * (1.36)**	**5.75 ** (1.70)**	N/A
Sexual orientation	.51 (.70)	−.56 (.76)	N/A
External shame in THBE	1.08 (.95)	.25 (.97)	1.72 (3.74)
Internal shame in THBE	**1.78 * (.83)**	**3.14 *** (.83)**	**9.23 ** (3.26)**
Compassionate engagement	.36 (.23)	**.55 ** (.23)**	.81 (.88)
Compassionate actions	**−.63 ** (.18)**	**−1.06 *** (.18)**	**−2.19 ** (.70)**

Significant predictors are in bold; * *p* < .05; ** *p* < .005; *** *p* < .001.

## Data Availability

Not applicable.

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
