# Peer review of "Shame-Based Experiences of Homophobic Bullying and Mental Health: The Mediating Role of Self-Compassionate Actions"

_ijerph, 2022, doi:10.3390/ijerph192416866_

Round 1

Reviewer 1 Report

Dear authors,

I find the revised article on homophobic bullying very interesting. The topic has been dealt with in a correct and respectful way. It is a topic that is of interest to society and therefore to the scientific community. However, I would like to make some suggestions in certain sections of the research.

Title, abstract and keywords. They are in line with the topic developed and allow you to know the purpose of the research. The abstract is very well organised and adequate. It presents the background, methodology, results and conclusions in a clear and orderly manner. According to what is desirable for a scientific abstract.

Introduction. The wording favours a fluent and comprehensive reading of the document. Sufficient previous studies are provided, but none are sufficiently up-to-date. They are all very old and do not reflect situations or experiences of the last five years, as recommended in current research. I strongly recommend reviewing and updating the sources consulted.

The proposed objectives are operational and adequately described. Figure 1 is timely and links well with the introduction and objectives.

Participants. The information on the selection of participants described in the "procedure" section should go in this section.

Table 1 gives a broader overview of the participants. This table is timely and necessary, as it does not repeat information.

Procedure. Adequate and sufficient description.

Instrument. Each of the scales used is described, reflecting the reliability values. In addition, the authors of the instruments are cited.

Statistical analysis. Appropriate analyses carried out with SPSS and AMOS are proposed.

Results. Descriptive statistics, correlations and regressions are analysed. Have normality tests been carried out? If so, this should be reflected in the text.

Analyses are shown in a clear and orderly manner. Self-compassion and its components presented a negative correlation with all the remaining variables; that is, higher levels of self-compassion were associated with lower levels of (external and internal) shame felt in THBE, anxiety, and with depression and social anxiety symptoms.

Regarding gender identity, transgender individuals showed higher levels of external 315 shame felt in THBE than cisgender individuals. These results are interesting and are adequately discussed.

Transgender individuals also presented higher levels of anxiety and depression symptoms when compared to cisgender individuals. In self-compassion, the only difference found was across genders. Women showed lower levels of self-compassion when compared to men.

The results found for each of the variables analysed are discussed one by one. This allows for a fluent and comprehensive reading of the article presented. The results are related to previous current studies, which is to be welcomed, as this had not been the case in the "introduction" section.

With regard to limitations, the limitation of self-reporting should be taken into account.

Conclusions. This section is very brief. It may well be extended or merged with the previous section.

References. A multitude of previous research is presented, cited according to MDPI standards.

In general, the article presents very enriching results that are particularly useful for the scientific community and society in general. I only propose some improvements in terms of the timeliness of the references proposed in the introduction, in the results and in the limitations.

Author Response

We kindly appreciate all your comments and suggestions.

We found your suggestion about the introduction very relevant. We revised all the introduction considering our search of more recent literature about the topics of interest.

Additionally, we changed the information about the selection of participants from “procedure” to “participants”, as you suggested. Thank you for your accurate eye.

Mistakenly, we forgot to write the normality test. We added the Kolmogorov-Sminorv test in “statistical analyses” and reported the respective results in “normality, descriptive statistics and correlations between variables”.

We also thoughfully considered your suggestion of describing the association between variables further.

Finally, we merged the “conclusion” into the “discussion”, referring the limitation of self-report measures.

Thanks again for your attentive and kind review. 

Reviewer 2 Report

Thank you for the opportunity to review this manuscript. I have only a few noted weaknesses, that, if addressed would improve its readability.

Within the results section, lines 257-284 are somewhat challenging to read. Several of these results appear to be displayed in the figure above and are easier to see within a visual depiction than text. One alternative might be to provide a supplemental table with the additional statistics to the figure and/or to start a new paragraph at line 274.

Within the discussion section, line 298-300, authors suggest "self-compassion may be a transversal positive factor to buffer shame and psychopathology indicators across sexual orientations." - I believe they are correct and would recommend authors look into, and, if appropriate, cite research on self-compassion and psychological resiliency here (for example, see work by Kristin Neff, PhD).

Within the discussion section, lines 356-357 - Authors state that there were lower levels of self-compassion among women--since they use the minority stress framework in explaining LGBTQ+ psychological stress -- it might be appropriate to consider (and cite papers) about social forces like patriarchy, misogyny, etc. that societally influence women to internalize less self-compassion.

Conclusions - "internal shame should be a target in clinical practice with this population" - I believe that internal shame is likely a target in clinical practice with MOST populations (inclusive of this one) as this is often a key factor in driving negative self-talk and maintaining anxiety/depression in clinical populations. However, I do agree, that findings support specifically targeting internalized shame through compassionate actions is interesting.

Author Response

We kindly appreciate all your comments and suggestions.

We appropriated your suggestion, separating the paragraphs to facilitate reading.

Regarding the resilience topic, we added it to the clinical implications. We considered the other proposed topic (about less compassionate actions in women’s social reflection) very relevant and we did some changes about it: we added some ideas about the masculine dominant culture that leads women to move away from fierce component of self-compassion. Thank you for your acumen.

Thanks again for this thoughtful review.